# Synthesis, Characterization, Computational and Biological Activity of Some Schiff Bases and Their Fe, Cu and Zn Complexes

**Tareq M. A. Al-Shboul [1], Mohammad El-khateeb [2], Zaid H. Obeidat [3], Taher S. Ababneh [3], Suha S. Al-Tarawneh [1], Mazhar S. Al Zoubi [4], Walhan Alshaer [5], Anas Abu Seni [3], Taqwa Qasem [4], Hayato Moriyama [6], Yukihiro Yoshida [6], Hiroshi Kitagawa [6] and Taghreed M. A. Jazzazi [3,*]**

1 Department of Chemistry and Chemical Technology, Tafila Technical University, Tafila 66110, Jordan; tareq92@hotmail.com (T.M.A.A.-S.); s.tarawneh@ttu.edu.jo (S.S.A.-T.)
2 Chemistry Department, Jordan University of Science and Technology, Irbid 22110, Jordan; kateeb@just.edu.jo
3 Department of Chemistry, Yarmouk University, Irbid 21163, Jordan; 2017104005@ses.yu.edu.jo (Z.H.O.); tababneh@yu.edu.jo (T.S.A.); anassos1982@hotmail.com (A.A.S.)
4 Department of Basic Medical Sciences, Yarmouk University, Irbid 21163, Jordan; mszoubi@yu.edu.jo (M.S.A.Z.); takwaqasem95@gmail.com (T.Q.)
5 Cell Therapy Center, The University of Jordan, Amman 11942, Jordan; walhan.alshaer@ju.edu.jo
6 Division of Chemistry, Graduate School of Science, Kyoto University, Kitashirakawa-Oiwakecho, Sakyo-ku, Kyoto 606-8502, Japan; moriyama.hayato.54m@ssc.kuchem.kyoto-u.ac.jp (H.M.); yoshiday@ssc.kuchem.kyoto-u.ac.jp (Y.Y.); kitagawa-kukem@googlegroups.com (H.K.)
* Correspondence: taghreed.j@yu.edu.jo

**Abstract:** Four new symmetrical Schiff bases derived from 2,2′-diamino-6,6′-dibromo-4,4′-dimethyl-1,1′-biphenyl or 2,2′-diamino-4,4′-dimethyl-1,1′-biphenyl, and 3,5-dichloro- or 5-nitro-salicylaldehyde, were synthesized and reacted with copper-, iron- and zinc-acetate, producing the corresponding complexes. The Schiff bases and their metal complexes were characterized by $^1$H-, $^{13}$C-NMR, IR and UV-Vis spectroscopy and elemental analysis. The structures of one Schiff base and the two zinc complexes were resolved by X-ray structure determination. Density functional theory (DFT) calculations at the B3LYP/6-31G(d) level of the latter compounds were carried out to optimize and examine their molecular geometries. The biomedical applications of the Schiff bases and their complexes were investigated as anticancer or antimicrobial agents.

**Keywords:** symmetrical Schiff base; metal complexes; crystal structure; anticancer; antimicrobial; DFT calculation

## 1. Introduction

Schiff bases are a well-documented class of ligands that are capable of bonding to almost all metals of the periodic table [1]. They are generally formed through a condensation reaction of carbonyl compounds with primary amines, in which mono-, di- or tri-functional amines lead to bi-, tri-, tetra- or poly-dentate Schiff bases. Symmetrical Schiff bases are formed when diamines are reacted with the same aldehydes or ketones in a 1:2 molar ratio. On the other hand, unsymmetrical Schiff bases may be obtained when two different aldehydes or ketones are reacted with diamines.

Due to the presence of N/O donor atoms in any Schiff base, they can coordinate to metals, leading to new metal complexes. Two main methods are used to synthesize Schiff base complexes. The first involves basic conditions, in which the Schiff base reacts with the metal ion in alcoholic or aqueous solutions [2]. The second methodology depends on a template reaction of primary amines, aldehydes or ketones to react simultaneously with metal ions [3]. This method is used to synthesize Schiff base complexes derived from macrocyclic and cyclic ligands.

Schiff bases and their complexes have versatile applications [4]. For instance, they are used in medicinal and pharmaceutical chemistry. In the medical sector, they are used

as anticancer, antitubercular, anti-inflammatory, antipyretic and analgesic agents [5–22]. Moreover, Schiff bases are used in dye synthesis [23], for antioxidative activity [24], immobilization of enzymes [25], cation carriers in potentiometric sensors [8,26] and as effective corrosion inhibitors [27]. Schiff bases and their metal complexes are also used in electrochemical sensors and chromatographic methods for detection with high selectivity and sensitivity [28,29]. The biological and medicinal activities of Schiff base metal complexes are greater than those of free Schiff bases [30–33]. Furthermore, divalent metal Schiff base complexes, especially those involving Fe(II), Ru(II) and Cu(II), have been widely used in catalysis for the oxidation of alcohols, cyclopropanation reactions and base hydrolysis of amino acid esters [34–42].

Motivated by all of the aforementioned applications, we are interested in the synthesis of new Schiff base ligands and their metal complexes. The reaction of $M(OAc)_2$ (M = Cu(II), Co(II) or Ni(II) with 2,2′-bis(2-hydroxybenzylideneamino)-4,4′-dimethyl-1,1′-biphenyl allowed the synthesis of their complexes. The ligands are in tetradentate binding mode via ONNO motif of the two phenolic oxygen atoms and two azomethine nitrogen atoms [43].

In our previous studies, new Schiff base ligands derived from 2,2′-diamino-4,4′-dimethyl-1,1′-biphenyl-salicylaldehyde were prepared and characterized. These ligands were then reacted with Cu(II), Mn(II) or Zn(II) acetate, forming tetra-coordinate metal complexes. The anticancerous and antiproliferative activities of one representative ligand and its metal complexes were reported [22]. Additionally, 4-thiazolidinone derivatives were prepared, and their biological activity, along with their metal complexes, was tested as a fungicide and found to have good activities against fungi [44]. The preparation and characterization of substituted 2,2′-bis(2-oxidobenzylideneamino)-4,4′-dimethyl-1,1′-biphenyl complexes of zinc, potassium and titanium were studied [45–47]. Recently, we undertook the evaluation and molecular modelling of bis-Schiff base derivatives as potential leads for the management of diabetes mellitus [48].

As an extension of our research in the area of Schiff base complexes, we report herein the synthesis of two new symmetrical Schiff bases by the condensation reactions of 2,2′-diamino-6,6′-dibromo-4,4′-dimethyl-1,1′-biphenyl and 2,2′-diamino-4,4′-dimethyl-1,1′-biphenyl with salicylaldehyde derivatives to produce the desired tetradentate Schiff base ligands. The copper, zinc and iron complexes of these new Schiff bases were obtained. To obtain a qualitative understanding of the structural characteristics and relative energies of the prepared Schiff bases and metal complexes, a density functional theory (DFT) computational analysis was carried out. Moreover, by certain biological testing, these Schiff bases and their complexes were demonstrated to be antimicrobial and anticancer agents.

## 2. Results and Discussion

### 2.1. Synthesis

Symmetrical Schiff bases were prepared via the condensation reaction of 2,2′-diamino-4,4′-dimethyl-1,1′-biphenyl or 2,2′-diamino-6,6′-dibromo-4,4′-dimethyl-1,1′-biphenyl with 3,5-dichloro- or 5-nitro-salicylaldehyde, as shown in Scheme 1. Copper(II), iron(II) and zinc(II) complexes of these Schiff bases were prepared by their reactions with metal acetate in a 1:1 ligand-to-metal molar ratio. A base (sodium methoxide or triethylamine) was added to subtract the acidic proton.

The metal complexes **Z1M**–**Z4M** were colored complexes and characterized by molar conductivity, UV-Vis, IR, [1]H- and [13]C-NMR spectra where applicable. The complexes had non-electrolytic behavior, evident by the very low (~0) molar conductivity, suggesting that the complexes were indeed neutral. The UV-Vis absorption spectra of the Schiff bases and their metal complexes were measured in DMSO solutions. The spectra of the Schiff bases displayed two bands (310–359 and 451–452 nm), which may be attributed to intraligand absorptions (π-π* and n-π) of the conjugated system and the azomethine group [49]. However, the metal complexes showed only one band in the range of 361–412 nm, which may be attributed to a metal-to-ligand charge transfer band.

**Scheme 1.** Synthesis of symmetrical Schiff bases **ZH1–ZH4** and their Cu, Fe and Zn complexes.

The $^1$H-NMR spectra of the free Schiff bases presented the phenolic signal (12.61–13.63 ppm). This peak disappeared in the spectra of the Zn(II) complexes. This proves that the two hydroxyl groups in the free Schiff base ligands lost their protons, and new bonds between the metal and the two oxygen atoms were formed. Moreover, the signal in the range of 8.04–8.58 ppm of the free Schiff bases that was attributed to the azomethine proton (CH=N) was shifted upfield in the zinc complexes' spectra, thereby supporting its bond formation with the Zn center. These $^1$H-NMR values for both the free ligands and zinc complexes were consistent with those published for similar compounds [22,45–47,50,51]. The $^{13}$C-NMR spectra of the free Schiff bases displayed peaks in the ranges of 21.23–21.39 ppm and 161.04–166.32 ppm, corresponding to the methyl and azomethine carbons, respectively. These peaks were shifted to 20.99–21.03 ppm and 168.72–175.62 ppm upon complexation to zinc. Although a small shift was observed for the methyl peak, a big shift was observed for the azomethine carbon, another indication of the N-bonding to zinc.

In the IR spectra of the Schiff bases, the characteristic bands in regions 3467–3450, 2900–3100, 1627–1613, 1476–1413 and 1356–1323 cm$^{-1}$ may be attributed to the absorption of $\nu$(O-H), $\nu$(C-H), $\nu$(C=N), $\nu$(C=C) and $\nu$(C-O), respectively. The $\nu$(C=N) band was shifted to a lower wavenumber in the corresponding spectra of the complexes. This indicated that this group was coordinated to the metal center. In addition, the disappearance of the weak and broad band of the hydroxyl group gave important evidence of the complexation of the phenolic groups. Moreover, two new bands, in the ranges of 400–450 and 500–550 cm$^{-1}$ due to M-N and M-O stretching, appeared in the metal complexes' spectra and represented another proof of O/N coordination to metals. These results are in good agreement with those previously published for similar complexes [22,52–56].

### 2.2. X-ray Structural Analysis

The single crystals of **ZH4** suitable for X-ray structure determination were grown from an EtOH solution. The structure is displayed in Figure 1, in which the C1–C15 length of the biphenyl moiety is 1.498(4) Å. The imine moieties had a C=N bond (C22-N3: 1.275(5) Å; C8-N1: 1.265(5) Å) slightly shorter than those of the unsubstituted analogue (1.283(1) Å) [47] and the analogous 3-methoxy substituted ligand (1.285(2) and 1.280(2) Å) [47]. This might be due to the presence of the electron-withdrawing nitro group. The C-N(nitro) bond lengths

of 1.451(5) and 1.423(5) Å were within a normal C-N single bond. The C-O distances (1.325(5) and 1.320(5) Å) were similar to those of the unsubstituted analogue (1.354(2) Å) and the analogous 3-methoxy substituted ligand (1.353(7) Å). However, the C-Br bond distances (1.896(3) and 1.893(3) Å) were within the same range observed for both the unsubstituted and MeO-substituted analogues [46,47].

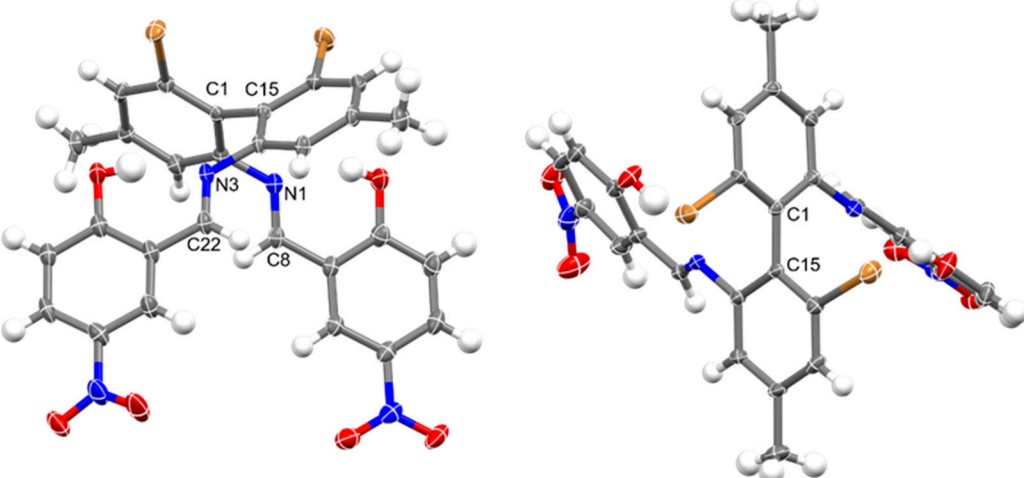

**Figure 1.** Molecular structure of **ZH4** determined by X-ray diffraction at 100 K. White: H, gray: C, blue: N, red: O, and orange: Br.

Samples of complexes **Z1Zn** and **Z2Zn** were recrystallized from the THF/pentane and $CH_2Cl_2$/MeOH solvent mixtures, respectively. **Z1Zn** was in a monoclinic system and contained THF as a solvent molecule. **Z2Zn** was in a triclinic system, where the contribution of highly disordered solvent molecules (42 $e^-$ in a unit cell) was removed using the PLATON/SQUEEZE [57]. The structures of these two complexes, with the atomic numbering schemes, are presented in Figures 2 and 3, respectively. The unit cell had two molecules, which were crystallographically independent. Because the molecular structures were quite similar to each other, only one molecule is discussed hereafter. The Zn atom, in each molecule, was penta-coordinated with two oxygen atoms from the deprotonated hydroxyl groups, as well as two nitrogen atoms from the azomethine groups, in addition to a water molecule occupying the apical position. The Zn-O bond lengths (**Z1Zn**: 2.027(3) and 1.966(3) Å; **Z2Zn**: 1.9496(18) and 1.9740(18) Å) were comparable to each other and to those found for bis(2-oxidobenzylideneamino)-4,4′-dimethyl-1,1′-biphenylzinc [46]. The Zn-N bond distances (**Z1Zn**: 2.048(3) and 2.107(3) Å; **Z2Zn**: 2.140(2) and 2.039(2) Å) were slightly longer than the Zn-O bonds within the same molecules. The Zn-O (of $H_2O$ molecules) bond distances (**Z1Zn**: 2.082(3) Å; **Z2Zn**: 2.1966(18) Å) were longer than the other Zn-O bond lengths within these molecules. These bond distances were very comparable to those reported for similar Zn complexes [46]. The geometry around the Zn center of **Z1Zn** was a distorted trigonal bipyramid (index of the degree of trigonality $\tau$ = 0.78 [58]), with the O1 and N2 in the axial positions (O1-Zn-N2 = 172.81(13)°), while N1, O2 and O3 occupied the equatorial places (O2-Zn-N1= 125.96(14)°; O2-Zn-O3 = 109.26(13)°; O3-Zn-N1= 124.75(13)°). This geometry was similar to that reported for bis(2-oxidobenzylideneamino)-4,4′-dimethyl-1,1′-biphenylzinc; however, in this case, one of the N atoms and the O of the coordinated methanol occupy the axial position [46]. The deviation of **Z2Zn** from the trigonal bipyramid was higher than that of **Z1Zn** ($\tau$ = 0.16). Two bond angles (O1-Zn-N2 = 159.44(8)°; O7-Zn-N1 = 146.80(8)°) were much higher than the equatorial angle of 120 degrees, while the other two angles (O2-Zn-N1 = 114.63(8)°; O1-Zn-O2 = 109.04(7)°) were slightly less than 120 degrees. This implied that both O2 and N1 were within the equatorial position in addition to O1, with greater deviation. This left O7 and N2 to most probably have axial positions.

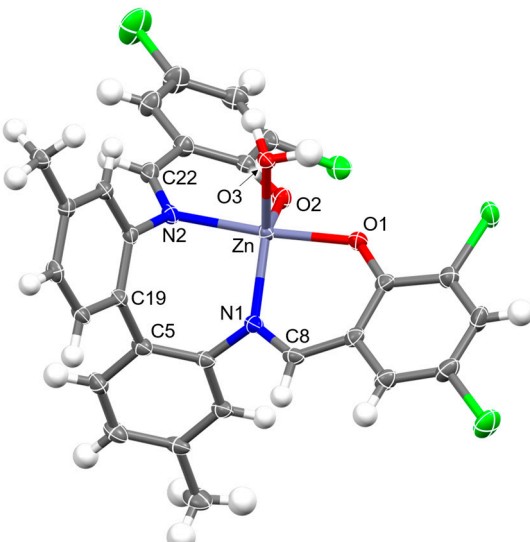

**Figure 2.** Molecular structure of **Z1Zn** determined by X-ray diffraction at 100 K. The ellipsoids are drawn with a 50% probability. White: H, gray: C, blue: N, red: O, light green: Cl, and blue-gray: Zn.

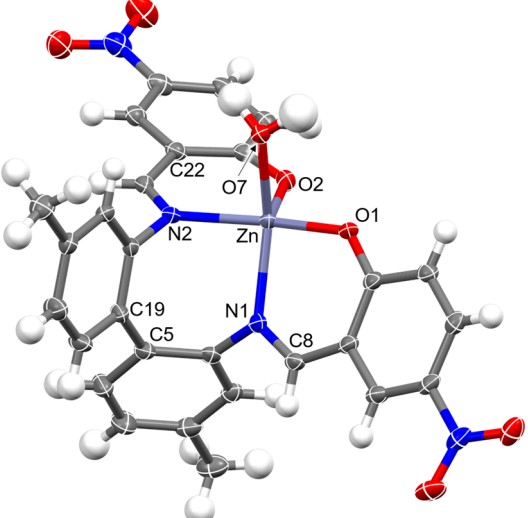

**Figure 3.** Molecular structure of **Z2Zn** determined by X-ray diffraction at 100 K. The ellipsoids are drawn with a 50% probability. White: H, gray: C, blue: N, red: O, and blue-gray: Zn.

*2.3. Computational Study*

The optimized geometry of **ZH4** is depicted in Figure 4. The selected structural parameters for the optimized ligand, along with those determined experimentally, are listed in Table 1.

The calculated C1–C15 bond length (1.492 Å) between the aryl units of the biphenyl backbone, which was estimated to be 1.498(4) Å by the crystallographic study, featured a typical single-bond value, thus ruling out interactions between the π-systems of these moieties.

This value was close to the calculated average value of 1.494 Å, for a series of twelve comparable complexes bearing the biphenyl backbone around the coordination sphere [18].

The aryl moieties of the biphenyl backbone were twisted with angles between the planes by 87.31°(exp. 83.89°). The large torsion angle was attributed to the steric requirements enforced by bromine substituents. This was consistent with the experimentally reported aryl–aryl dihedral angle of 80° for a similar dibromo-substituted biphenyl ligand [59]. Smaller torsion angles (between 21° and 68°) for the halide-free analogues, where there was flexible twisting between the two biphenyl planes, were observed [46]. The optimized geometries of **Z1Zn** and **Z2Zn** are elucidated in Figure 5. The results indicated

that the monoligated complexes exhibited a penta-coordinated zinc ion with distorted trigonal bipyramidal geometry. The computed $H_2O\cdots M$ bond was 2.247 Å (exp. 2.082(3) Å) and 2.238 Å (exp. 2.1966(18) Å) for the **Z1Zn** and **Z2Zn** complexes, respectively. Additionally, the calculated Zn-O bond lengths around the Zn ion were [**Z1Zn**: 1.970, 1.947 Å (exp. 2.027(3) and 1.966(3) Å); **Z2Zn**: 1.949, 1.983 Å (exp. 1.9496(18) and 1.9740(18) Å)], while the calculated Zn-N bond lengths were [Z1Zn: 2.048, 2.102 Å, (exp. 2.048(3) and 2.107(3) Å); **Z2Zn**: 2.087, 2.057, Å (exp. 2.140(2) and 2.039(2) Å)].

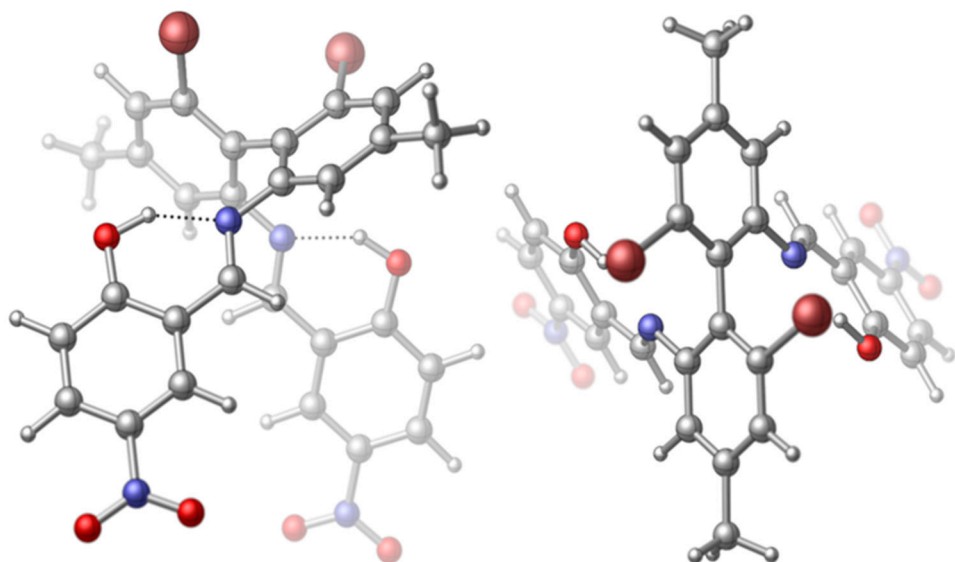

**Figure 4.** Different views of the optimized ground-state geometry for the **ZH4** ligand at the B3LYP/6-31G(d) level of theory. (O: red, N: blue, C: gray, and Br: brick). Dotted line for O–H⋯N hydrogen bonding.

**Table 1.** Selected calculated and experimental bond lengths (Å) and angles (°) of **ZH4**.

| Bond (Å) | Calc. | Exp. | Angle (°) | Calc. | Exp. |
|----------|-------|------|-----------|-------|------|
| C1–C15 | 1.492 | 1.498 | O-N-O | 124.53, 124.53 | 122.95, 123.04 |
| C=N | 1.290, 1.290 | 1.265, 1.275 | C-O-H | 107.92, 107.92 | 100.31, 106.76 |
| C-O | 1.332, 1.332 | 1.325, 1.320 | C-N=C | 121.04, 121.03 | 121.20, 120.86 |
| C-Br | 1.918, 1.918 | 1.896, 1.893 | C-C=N | 121.89, 121.90 | 120.41, 120.38 |
| C-N(imine) | 1.408, 1.408 | 1.424, 1.427 | Br-C-CH | 117.56, 117.56 | 118.09, 118.30 |
| C-N(nitro) | 1.461, 1.461 | 1.453, 1.451 | Br-C-C | 120.05, 120.05 | 119.28, 119.05 |
| C-CH$_3$ | 1.510, 1.510 | 1.515, 1.509 | | | |
| N⋯H | 1.742, 1.742 | 1.804, 1.896 | | | |

Such slight structural variations between the experimental and calculated values were attributed to the experimental data being acquired for crystalline materials with lattice interactions, such as packing effects and intermolecular van der Waals forces, whereas the calculated values corresponded to an isolated molecule in the gas phase.

The calculated bond length between the aryl units of the biphenyl backbone was 1.489 Å (exp. 1.489 Å) and 1.488 Å (exp. 1.491 Å) for the **Z1Zn** and **Z2Zn** complexes, respectively, alluding to a typical single bond and excluding π-interactions between the aryl units. The calculated twist angle between the planes of the aryl moieties of the biphenyl backbone was 59.39°(exp. 60.61°) and 59.19°(exp. 55.81°) for the **Z1Zn** and **Z2Zn** complexes, respectively. Such large torsion angles were attributed to the steric and electronic requirements enforced within the coordination sphere.

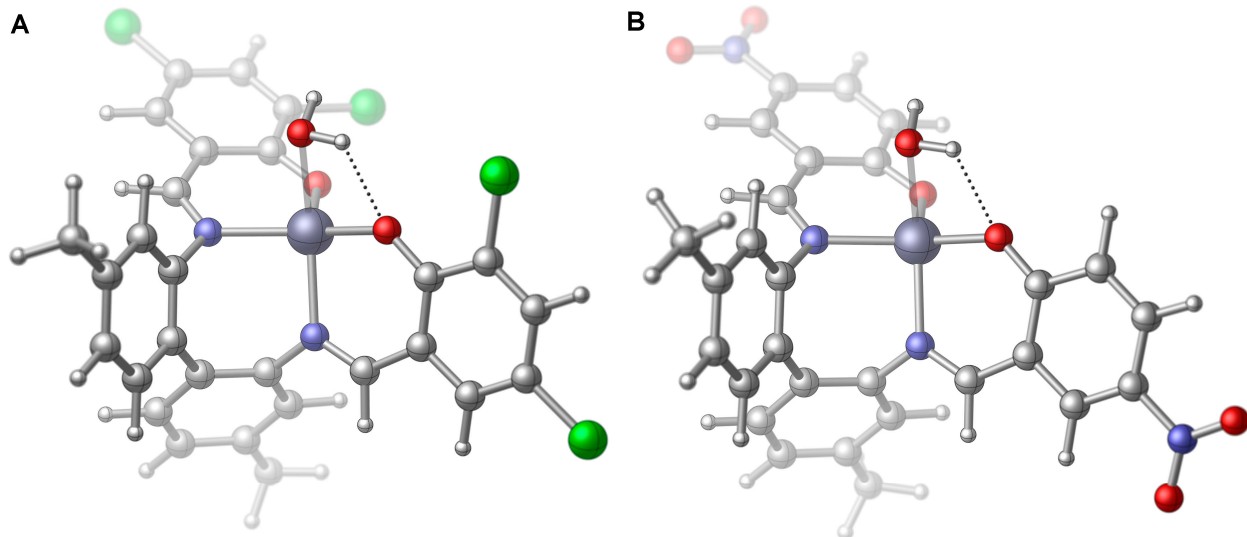

**Figure 5.** The optimized ground-state geometries of **Z1Zn** (**A**) and **Z2Zn** (**B**) complexes at the B3LYP/6-31G(d) level of theory. (O: red, N: blue, C: gray, and Cl: green). Dotted line for O–H···O hydrogen bonding.

## 2.4. Antimicrobial and Anticancer Assays

The results showed significant antimicrobial activity for both the Schiff bases and their metal complexes against Gram-positive bacterial strains *(Micrococcus luteus* and *Staphylococcus aureus)*, which were demonstrated as a zone of inhibition (Table 2, Figure 6). These compounds showed different activity towards the two bacteria types. For instance, **ZH2**, **ZH3**, **ZH4** and their zinc complexes showed antimicrobial activity against *Micrococcus luteu*s and *Staphylococcus aureus*, while **Z2Fe**, **Z4Fe** and **Z4Cu** showed antimicrobial activity against *Staphylococcus aureus* only. The selectivity of these compounds raises an interesting concern about the exact mechanism of these compounds as antimicrobial agents. The Schiff bases and certain complexes showed limited antimicrobial activity on Gram-negative bacteria (*Escherichia coli*).

**Table 2.** The antimicrobial activity as shown by the inhibition area in the tested compounds.

| Tested Compounds (10 mg mL$^{-1}$) Inhibition Zone (mm) | *Micrococcus luteus* (ATCC 934) | *Staphylococcus aureus* (ATCC 29213) | *Escherichia coli* (ATCC 25922) |
|---|---|---|---|
| ZH1 | - | - | - |
| ZH2 | 10 | 12 | - |
| ZH3 | 14 | 8 | - |
| ZH4 | 36 | 22 | - |
| [Z1Zn] | - | - | - |
| [Z2Zn] | 15 | 21 | - |
| [Z3Zn] | 25 | 18 | - |
| [Z4Zn] | 15 | 25 | - |
| [Z1Fe] | - | - | - |
| [Z2Fe] | - | 14 | - |
| [Z3Fe] | - | - | - |
| [Z4Fe] | - | 20 | - |
| [Z1Cu] | - | - | 10 |
| [Z2Cu] | - | - | - |
| [Z3Cu] | - | - | 20 |
| [Z4Cu] | - | 11 | - |
| Amoxicillin | 25 | 35 | 10 |

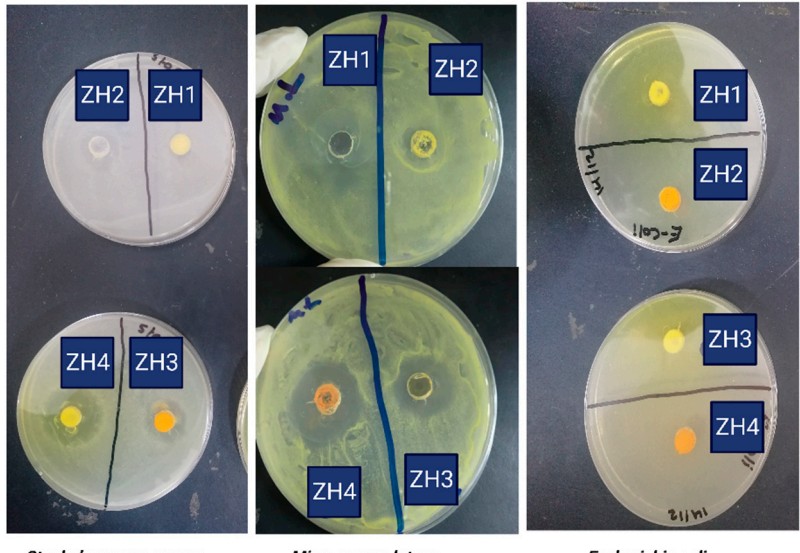

**Figure 6.** A representative bacterial culture showing the inhibition zone in Gram-positive (*Staphylococcus aureus* and *Micrococcus luteus*) bacteria by compounds **ZH1**, **ZH2**, **ZH3** and **ZH4** ((**left**) and (**middle**)) compared to lack of antimicrobial activity against Gram-negative bacteria (*Escherichia coli*) (**right**).

The $IC_{50}$ of the Schiff bases and their metal complexes showed a variable reduction in viability of the tested cancer cell lines compared to the normal fibroblasts. For instance, the $IC_{50}$ of **ZH1** showed significant inhibition of cancer cell line growth compared to the HDF cell line. In addition, the $IC_{50}$ of the tested compounds showed a selective anticancer effect against MCF7 compared to the A549 and HDF cell lines, as shown in Table 3. **ZH1**, its zinc complexes and **Z2Fe** showed selective anticancer effects against the MCF7 cell line. Interestingly, **Z2Cu** showed the lowest $IC_{50}$ values against all cell lines.

**Table 3.** $IC_{50}$ values (µg/mL) of the tested compounds after the cell viability assay (MTT), showing variable responses of the treated cell lines with different specificity.

| Cell Lines | ZH1 | ZH2 | ZH3 | ZH4 | Z1Zn | Z1Cu | Z2Zn | Z2Fe | Z2Cu | Z3Zn | Z3Cu | Z4Zn | Z4Fe | Z4Cu | Doxorubicin |
|---|---|---|---|---|---|---|---|---|---|---|---|---|---|---|---|
| A549 | 25.2 | 320.3 | 41.0 | 62.17 | 224.9 | 43.7 | 81.2 | 118.1 | 4.0 | 199.4 | 130.0 | 25.2 | 68.6 | 34.7 | 0.15 |
| MCF7 | 20.5 | 231.7 | 77.5 | Not covered | 17.3 | 10.4 | 25.2 | 41.4 | 1.9 | 17.7 | 172.0 | 14.1 | 36.5 | 19.2 | 0.03 |
| Fibroblasts | 46.8 | 291.8 | 71.1 | 69.2 | 68.2 | 8.5 | 68.7 | 134.1 | 1.5 | 27.6 | 29.2 | 28.3 | 16.7 | 10.4 | 0.37 |

## 3. Experiment

### 3.1. Materials and Methods

Elemental analyses (C, H and N) were measured at the Institute of Organic and Macromolecular Chemistry of the Friedrich Schiller University in Jena, Germany. Infrared spectra were recorded with a Bruker FT-IR-4100 spectrometer (Bremen, Germany). $^1$H- and $^{13}$C-NMR spectra were recorded on a Bruker AC 400 MHz spectrometer. Chemical shifts were reported in ppm relative to TMS as internal standards. Melting points were recorded on a melting point apparatus, SMP3. Electronic absorption spectra were measured on a PS-2600 Pasco spectrophotometer (Saarbrucken, Germany) in DMSO solvent using $8 \times 10^{-5}$ M solutions. All reagents were obtained from Merck, Acros or Sigma Aldrich (Taufkirchen, Germany) and used without further purification. The starting amines, 2,2′-diamino-4,4′-dimethyl-6,6′-dibromobiphenyl [29] and halide-free analogue [47], were prepared according to literature procedures. The theoretical modelling of Schiff base ligands was performed using Wave function Spartan'18 Parallel Suite on a desktop computer with core i725.

### 3.2. General Procedure for the Preparation of Ligands (ZH1–ZH4)

2,2′-Diamino-4,4′-dimethyl-6,6′-dibromobiphenyl or 2,2′-diamino-4,4′-dimethyl-1,1′-biphenyl (2.35 mmol) and salicylaldehyde derivatives (4.7 mmol) in 10 mL of absolute ethanol was refluxed for 5 h. During this time, the corresponding Schiff base was precipitated, collected, washed with ethanol and stripped to dryness. The solid was recrystallized from a $CH_2Cl_2$/methanol mixture to give pure crystalline compounds.

**2,2′-Bis(3,5-dichlorosalicylideneamino)-4,4′-dimethyl-1,1′-biphenyl (ZH1):**

Yellow (55%). M.P. = 233–235 °C. $^1$H-NMR (400 MHz, $CDCl_3$): 13.19 (2 H, s, OH); 8.18 (2 H, s, CH=N); 6.76–7.28 (10 H, m, Ar-H); 2.37 (6 H, s, $CH_3$). $^{13}$C-NMR (400 MHz, $CDCl_3$): 161.04 (C=N); 119.17–155.22 (Ar); 21.30 ($CH_3$). Anal. Calc. for $C_{28}H_{20}Cl_4N_2O_2$: C, 60.24; H, 3.61; N, 5.02%. Found: C, 60.13; H, 3.59; N, 5.00%. (IR, $cm^{-1}$, KBr): $\upsilon$(O-H) = 3467 (br,w); $\upsilon$(C-H) = 2921 (w); $\upsilon$(C=N) = 1619 (s); $\upsilon$(C=C) = 1448 (m); $\upsilon$(C-O) = 1352 (s); $\upsilon$(C-Cl) = 775. UV-Vis in $CH_2Cl_2$: $\lambda_{max}$(nm) ($\varepsilon_{max}$, $M^{-1}cm^{-1}$): 359 (2.37 × 10$^3$), 452 (3.75 × 10$^1$).

**2,2′-Bis(5-nitrosalicylideneamino)-4,4′-dimethyl-1,1′-biphenyl (ZH2):**

Orange (63%). M.P. = 197–199 °C. $^1$H-NMR (400 MHz, $CDCl_3$): 13.62 (2 H, s, OH); 8.50 (2 H, s, CH=N); 6.83–8.10 (12 H, m, Ar-H); 2.42 (6 H, s, $CH_3$). $^{13}$C-NMR (400 MHz, $CDCl_3$): 166.37 (C=N); 118.15–160.33 (Ar); 21.39 (CH3). Anal. Calc. for $C_{28}H_{22}N_4O_6$: C, 65.88; H, 4.34; N, 10.97%. Found: C, 65.86; H, 4.22; N, 10.87%. (IR, $cm^{-1}$, KBr): $\upsilon$(O-H) = 3461 (br,w); $\upsilon$(C-H) = 2923 (w); $\upsilon$(C=N) = 1917 (s); $\upsilon$(C=C) = 1476 (m); $\upsilon$(C-O) = 1340 (s); $\upsilon$($NO_2$) = 1385, 1574. UV-Vis in $CH_2Cl_2$: $\lambda_{max}$(nm) ($\varepsilon_{max}$, $M^{-1}cm^{-1}$): 351 (2.31 × 10$^3$), 452 (1.76 × 10$^2$).

**3,5-Dichlorosalicylideneamino-6,6′-dibromo-4,4′-dimethyl-1,1′-biphenyl (ZH3):**

Orange (79%). M.P. = 249–250 °C. $^1$H-NMR (400 MHz, $CDCl_3$): 12.61 (2 H, s, OH); 8.36 (2H, s, CH=N); 6.92–7.41 (8 H, m, Ar-H); 2.38 (6 H, s, $CH_3$). $^{13}$C-NMR (400 MHz, $CDCl_3$): 161.58 (C=N); 118.28–155.51 (Ar); 21.23 ($CH_3$). Anal. Calc. for $C_{28}H_{18}Cl_4Br_2N_2O_2$: C, 46.96; H, 2.53; N, 3.91%. Found: C, 46.92; H, 2.50; N, 3.89%. (IR, $cm^{-1}$, KBr): $\upsilon$(O-H) = 3465 (br,w); $\upsilon$(C-H) = 2923 (w); $\upsilon$(C=N) = 1613 (s); $\upsilon$(C=C) = 1452 (m); $\upsilon$(C-O) = 1354 (s); $\upsilon$(C-Br) = 669; $\upsilon$(C-Cl) = 742. UV-Vis in $CH_2Cl_2$: $\lambda_{max}$(nm) ($\varepsilon_{max}$, $M^{-1}cm^{-1}$): 357 (5.85 × 10$^3$), 459 (7.62 × 10$^1$).

**5-Nitrosalicylideneamino-6,6′-dibromo-4,4′-dimethyl-1,1′-biphenyl (ZH4):**

Yellow (82%). M.P. = 244–245 °C. $^1$H-NMR (400 MHz, $CDCl_3$): 13.13 (2 H, s, OH); 8.58 (2 H, s, CH=N); 6.83–8.25 (10 H, m, Ar-H); 2.41 (6 H, s, $CH_3$). $^{13}$C-NMR (400 MHz, $CDCl_3$): 166.32 (C=N), 117.76–161.11 (Ar); 21.31 ($CH_3$). Anal. Calc. for $C_{28}H_{20}Br_2N_4O_6$: C, 50.32; H, 3.02; N, 8.38%. Found: C, 50.32; H, 2.99; N, 8.37%. (IR, $cm^{-1}$, KBr): $\upsilon$(O-H) = 3450 (br,w); $\upsilon$(C-H) = 2940 (w); $\upsilon$(C=N) = 1619 (s); $\upsilon$(C=C) = 1413 (m); $\upsilon$(C-O) = 1338 (s); $\upsilon$(C-Br) = 646; $\upsilon$($NO_2$) = 1393, 1578. UV-Vis in $CH_2Cl_2$: $\lambda_{max}$(nm) ($\varepsilon_{max}$, $M^{-1}cm^{-1}$): 310 (5.45 × 10$^3$), 451 (1.85 × 10$^2$).

### 3.3. General Procedure for the Synthesis of Cu, Fe and Zn Complexes

To a solution of 1.30 mmol Schiff base in 10 mL of absolute ethanol, 1.30 mmol of metal(II) acetate hydrate dissolved in 3 mL of absolute ethanol was added dropwise at room temperature under inert atmosphere. The reaction mixture was refluxed for 6 h. During this time, the corresponding complex was precipitated, collected by filtration, washed with cold ethanol and dried under a vacuum.

**Z1Zn:** Yellow (60%). M.P. = 270–272 °C. $^1$H-NMR (400 MHz, $CDCl_3$): 8.16 (2 H, s, CH=N); 6.91–7.49 (10 H, m, Ar-H); 2.42 (6 H, s, $CH_3$). $^{13}$C-NMR (400 MHz, $CDCl_3$): 170.32 (C=N), 118.41–164.08 (Ar); 21.01 ($CH_3$). Anal. Calc. for $ZnC_{28}H_{18}Cl_4N_2O_2$: C, 54.10; H, 2.92; N, 4.51%. Found: C, 54.08; H, 2.90; N, 4.49%. (IR, $cm^{-1}$, KBr): $\upsilon$(C-H) = 2923 (w); $\upsilon$(C=N) = 1605 (s); $\upsilon$(C=C) = 1438 (m); $\upsilon$(C-O) = 1317 (s); $\upsilon$(Zn-N) = 430 (w); $\upsilon$(Zn-O) = 542 (w); $\upsilon$(C-Cl) = 764. UV-Vis in $CH_2Cl_2$: $\lambda_{max}$(nm) ($\varepsilon_{max}$, $M^{-1}cm^{-1}$): 396 (2.21 × 10$^3$).

**Z2Zn:** Yellow (72%). M.P. = 250–253 °C. $^1$H-NMR (400 MHz, $CDCl_3$): 8.28 (2 H, s, CH=N); 6.78–8.18 (12 H, m, Ar-H); 2.35 (6 H, s, $CH_3$). $^{13}$C-NMR (400 MHz, $CDCl_3$): 175.39 (C=N), 116.92–170.82 (Ar); 21.03 ($CH_3$). Anal. Calc. for $ZnC_{28}H_{20}N_4O_6$: C, 58.60; H, 5.51; N, 9.76%. Found: C, 58.57; H, 5.48; N, 9.73%. (IR, $cm^{-1}$, KBr): $\upsilon$(C-H) = 2923 (w); $\upsilon$(C=N)

= 1605 (s); $\upsilon$(C=C) = 1472 (m); $\upsilon$(C-O) = 1313 (s); $\upsilon$(Zn-N) = 416 (w); $\upsilon$(Zn-O) = 507 (w); $\upsilon$(NO$_2$) = 1391, 1550. UV-Vis in CH$_2$Cl$_2$: $\lambda_{max}$(nm) ($\varepsilon_{max}$, M$^{-1}$cm$^{-1}$): 384 (5.80 × 10$^3$).

**Z3Zn:** Yellow (49%). M.P. > 293 °C. $^1$H-NMR (400 MHz, CDCl$_3$): 8.10 (2 H, s, CH=N); 6.76–7.34 (8 H, m, Ar-H); 2.30 (6 H, s, CH$_3$). $^{13}$C-NMR (400 MHz, CDCl$_3$): 168.72 (C=N), 118.16–163.89 (Ar); 20.99 (CH$_3$). Anal. Calc. for ZnC$_{28}$H$_{16}$Cl$_4$Br$_2$N$_2$O$_2$: C, 43.14; H, 2.07; N, 3.59%. Found: C, 42.99; H, 1.98; N, 3.54%. (IR, cm$^{-1}$, KBr): $\upsilon$(C-H) = 2921 (w); $\upsilon$(C=N) = 1611 (s); $\upsilon$(C=C) = 1440 (m); $\upsilon$(C-O) = 1315 (s); $\upsilon$(Zn-N) = 428 (w); $\upsilon$(Zn-O) = 550 (w); $\upsilon$(C-Br) = 673; $\upsilon$(C-Cl) = 758. UV-Vis in CH$_2$Cl$_2$: $\lambda_{max}$(nm) ($\varepsilon_{max}$, M$^{-1}$cm$^{-1}$): 400 (2.95 × 10$^3$).

**Z4Zn:** Yellow (74%). M.P. = 270–273 °C. $^1$H-NMR (400 MHz, CDCl$_3$): 8.35 (2 H, s, CH=N); 6.74–8.19 (10 H, m, Ar-H); 2.34 (6 H, s, CH$_3$). $^{13}$C-NMR (400 MHz, CDCl$_3$): 175.62 (C=N), 116.69–169.62 (Ar); 21.03 (CH$_3$). Anal. Calc. for ZnC$_{28}$H$_{18}$Br$_2$N$_4$O$_6$: C, 45.96; H, 2.48; N, 7.66%. Found: C, 45.93; H, 2.44; N, 7.62%. (IR, cm$^{-1}$, KBr): $\upsilon$(C-H) = 2921 (w); $\upsilon$(C=N) = 1609 (s); $\upsilon$(C=C) = 1385 (m); $\upsilon$(C-O) = 1315 (s); $\upsilon$(Zn-N) = 416 (w); $\upsilon$(Zn-O) = 491 (w); $\upsilon$(C-Br) = 650; $\upsilon$(NO$_2$) = 1385, 1546. UV-Vis in CH$_2$Cl$_2$: $\lambda_{max}$(nm) ($\varepsilon_{max}$, M$^{-1}$cm$^{-1}$): 382 (2.24 × 10$^3$).

**Z1Fe:** Red (63%). M.P. = 260–261 °C. Anal. Calc. for FeC$_{28}$H$_{18}$Cl$_4$N$_2$O$_2$: C, 54.94; H, 4.34; N, 4.58%. Found: C, 51.92; H, 4.33; N, 4.54%. (IR, cm$^{-1}$, KBr): $\upsilon$(C-H) = 2923 (w); $\upsilon$(C=N) = 1605 (s); $\upsilon$(C=C) = 1436(m); $\upsilon$(C-O) = 1317 (s); $\upsilon$(Fe-N) = 442 (w); $\upsilon$(Fe-O) = 491 (w); $\upsilon$(C-Cl) = 771. UV-Vis in CH$_2$Cl$_2$: $\lambda_{max}$(nm) ($\varepsilon_{max}$, M$^{-1}$cm$^{-1}$): 363 (2.31 × 10$^3$).

**Z2Fe:** Red (66%). M.P. > 290 °C. Anal. Calc. for FeC$_{28}$H$_{20}$N$_4$O$_6$: C, 59.59; H, 3.57; N, 9.95%. Found: C, 59.53; H, 3.53; N, 9.95%. (IR, cm$_{-1}$, KBr): $\upsilon$(C-H) = 2921 (w); $\upsilon$(C=N) = 1607 (s); $\upsilon$(C=C) = 1468 (m); $\upsilon$(C-O) = 1315 (s); $\upsilon$(Fe-N) = 416 (w); $\upsilon$(Fe-O) = 504 (w); $\upsilon$(NO$_2$) = 1376, 1552. UV-Vis in CH$_2$Cl$_2$: $\lambda_{max}$(nm) ($\varepsilon_{max}$, M$^{-1}$cm$^{-1}$): 361 (6.70 × 10$^3$).

**Z3Fe:** Red (60%). M.P. > 320 °C. Anal. Calc. for FeC$_{28}$H$_{16}$Cl$_4$Br$_2$N$_2$O$_2$: C, 43.68; H, 2.09; N, 3.64%. Found: C, 43.63; H, 1.97; N, 3.62%. (IR, cm$^{-1}$, KBr): $\upsilon$(C-H) = 2921 (w); $\upsilon$(C=N) = 1611 (s); $\upsilon$(C=C) = 1439 (m); $\upsilon$(C-O) = 1317 (s); $\upsilon$(Fe-N) = 450 (w); $\upsilon$(Fe-O) = 497 (w); $\upsilon$(C-Br) = 673; $\upsilon$(C-Cl) = 762. UV-Vis in CH$_2$Cl$_2$: $\lambda_{max}$(nm) ($\varepsilon_{max}$, M$^{-1}$cm$^{-1}$): 368 (2.24 × 10$^3$).

**Z4Fe:** Red (72%). M.P. > 289 °C Anal. Calc. for FeC$_{28}$H$_{18}$Br$_2$N$_4$O$_6$: C, 46.57; H, 2.51; N, 7.76%. Found: C, 46.55; H, 2.49; N, 7.74%. (IR, cm-1, KBr): $\upsilon$(C-H) = 2923 (w); $\upsilon$(C=N) = 1609 (s); $\upsilon$(C=C) = 1384 (m); $\upsilon$(C-O) = 1315 (s); $\upsilon$(Fe-N) = 407 (w); $\upsilon$(Fe-O) = 485 (w); $\upsilon$(C-Br) = 652; $\upsilon$(NO$_2$) = 1399, 1552. UV-Vis in CH$_2$Cl$_2$: $\lambda_{max}$(nm) ($\varepsilon_{max}$, M$^{-1}$cm$^{-1}$): 359 (5.64 × 10$^3$).

**Z1Cu:** Green (71%). M.P. > 250 °C. Anal. Calc. for CuC$_{28}$H$_{18}$Cl$_4$N$_2$O$_2$: C, 54.26; H, 2.93; N, 4.52%. Found: C, 54.18; H, 2.87; N, 4.47%. (IR, cm$^{-1}$, KBr): $\upsilon$(C-H) = 2914 (w); $\upsilon$(C=N) = 1601 (s); $\upsilon$(C=C) = 1434(m); $\upsilon$(C-O) = 1323 (s); $\upsilon$(Cu-N) = 416 (w); $\upsilon$(Cu-O) = 538 (w); $\upsilon$(C-Cl) = 764. UV-Vis in CH$_2$Cl$_2$: $\lambda_{max}$(nm) ($\varepsilon_{max}$, M$^{-1}$cm$^{-1}$): 403 (2.32 × 10$^3$).

**Z2Cu:** Green (69%). M.P. > 270 °C. Anal. Calc. for CuC$_{28}$H$_{20}$N$_4$O$_6$: C, 58.79; H, 3.52; N, 9.79%. Found: C, 58.78; H, 3.49; N, 9.68%. (IR, cm$^{-1}$, KBr): $\upsilon$(C-H) = 2925 (w); $\upsilon$(C=N) = 1605 (s); $\upsilon$(C=C) = 1466 (m); $\upsilon$(C-O) = 1315 (s); $\upsilon$(Cu-N) = 430 (w); $\upsilon$(Cu-O) = 518 (w); $\upsilon$(NO$_2$) = 1383, 1546. UV-Vis in CH$_2$Cl$_2$: $\lambda_{max}$(nm) ($\varepsilon_{max}$, M$^{-1}$cm$^{-1}$): 389 (4.45 × 10$^3$).

**Z3Cu:** Green (67%). M.P. > 282 °C. Anal. Calc. for CuC$_{28}$H$_{16}$Cl$_4$Br$_2$N$_2$O$_2$: C, 43.25; H, 2.07; N, 3.60%. Found: C, 43.19; H, 2.02; N, 3.61%. (IR, cm$^{-1}$, KBr): $\upsilon$(C-H) = 2916 (w); $\upsilon$(C=N) = 1605 (s); $\upsilon$(C=C) = 1434 (m); $\upsilon$(C-O) = 1317 (s); $\upsilon$(Cu-N) = 416 (w); $\upsilon$(Cu-O) = 548 (w); $\upsilon$(C-Br) = 677; $\upsilon$(C-Cl) = 756. UV-Vis in CH$_2$Cl$_2$: $\lambda_{max}$(nm) ($\varepsilon_{max}$, M$^{-1}$cm$^{-1}$): 412 (3.38 × 10$^3$).

**Z4Cu:** Green (74%). M.P. >303 °C Anal. Calc. for CuC$_{28}$H$_{18}$Br$_2$N$_4$O$_6$: C, 46.08; H, 2.49; N, 7.68%. Found: C, 46.05; H, 2.46; N, 7.66%. (IR, cm$^{-1}$, KBr): $\upsilon$(C-H) = 2927 (w); $\upsilon$(C=N) = 1607 (s); $\upsilon$(C=C) = 1378 (m); $\upsilon$(C-O) = 1317 (s); $\upsilon$(Cu-N) = 516 (w); $\upsilon$(Cu-O) = 548 (w); $\upsilon$(C-Br) = 652; $\upsilon$(NO$_2$) = 1378, 1546. UV-Vis in CH$_2$Cl$_2$: $\lambda_{max}$(nm) ($\varepsilon_{max}$, M$^{-1}$cm$^{-1}$): 388 (4.10 × 10$^3$).

### 3.4. Computational Study

The ground-state molecular geometries of **ZH4**, **Z1Zn** and **Z2Zn** were fully optimized in the gas phase, without any constraints at the B3LYP/6-31G(d) level of theory [60–62].

The initial geometries were extracted from the experimentally determined crystal structure. All electronic structure calculations were performed using the Spartan'18 package [63].

### 3.5. Antimicrobial Assay

For the evaluation of the antibacterial activity of all synthesized compounds, two Gram-positive bacteria and Gram-negative bacteria were used in the current study: *Micrococcus luteus* and *Staphylococcus aureus*, and *Escherichia coli*, respectively. The antimicrobial assay procedures were performed according to the recommendation of the National Committee for Clinical Laboratory Standards (NCCLS) [64–66]. Briefly, the synthesized compounds were dissolved in DMSO at a concentration of 10 mg/mL. After the inoculation of the bacterial strains in the culture media, 50 μL sized wells were generated in the agar media, followed by the addition of 50 μL of the tested compounds. After 24–48 h of incubation, the diameter of the inhibition zone was measured.

### 3.6. Anticancer Assay

#### 3.6.1. Cell Culture

The parental breast cancer cell line (MCF7), non-small-cell lung cancer cell line (A549) and human dermal fibroblasts cell line (HDF) were obtained from the American Type Culture Collection (ATCC, Manassas, VA, USA). MCF7 and A549 cells were cultured as an attached monolayer and maintained in RPMI 1640 medium (EuroClone, Milan, Italy) supplemented with 10% (*v/v*) heat-inactivated fetal bovine serum (FBS) (Euro-Clone), 1% penicillin-streptomycin (EuroClone) and 2 mM L-glutamine. MDA-MB-231 cells were cultured as an attached monolayer and maintained in MEM (EuroClone) supplemented with 10% (*v/v*) heat-inactivated fetal bovine serum (FBS) (EuroClone), 1% penicillin-streptomycin (EuroClone) and 2 mM L-glutamine. The HDF were cultured as an attached monolayer and maintained in DMEM (EuroClone) supplemented with 10% (*v/v*) heat-inactivated fetal bovine serum (FBS) (EuroClone), 1% penicillin-streptomycin (EuroClone) and 2 mM L-glutamine. All cells were incubated at 37 °C in a 5% $CO_2$ tissue culture incubator (Memmert, Schwabach, Germany).

#### 3.6.2. Cell Viability Assay (MTT)

To determine the $IC_{50}$ of the **ZH1–ZH4** and **Z1Zn–Z4Cu** compounds on the selected cell lines, an MTT assay was performed. Approximately ($8 \times 10^3$ cells/well) of MDA-MB-231, MCF7 and HDF cell lines were seeded into a 96-well plate (Corning, Burlington, VT, USA). All cell lines were treated with different concentrations of the **ZH1–ZH4** and **Z1Zn–Z4Cu** compounds, ranging from 0.5 to 500 μg/mL. Then, the cells were incubated at 37 °C in a 5% $CO_2$ incubator for 72 h, after which the old media was aspirated, and the MTT assay salt (Bioworld, Visalia, CA, USA) in 100 μL of fresh media was added to each well. Following that, the plates were incubated at 37 °C for 3 h, and then 50 μL of solubilization solution (DMSO) was added to each well to determine viability. The absorbance of the solution was measured at 560 nm using a Glomax plate reader (Promega, Madison, WI, USA).

### 3.7. X-ray Structure Determination of ZH4, Z1Zn and Z2Zn

The intensity data of the X-ray diffraction peaks were collected on a Rigaku XtaLAB (Japan) P200 diffractometer, using Mo Kα radiation ($\lambda$ = 0.71073 Å) fitted with SHELXTL for structure determination [67]. A direct method using SHELXS-2014 was employed to solve the structure, and Fourier transformation was carried out using SHELXL-2014, employing full-matrix least-square refinement calculations [67].

#### 3.7.1. Crystallographic Data for **ZH4**

$C_{28}H_{20}Br_2N_4O_6$, $M_r$ = 668.30 g mol$^{-1}$, yellow block, triclinic, space group *P-1 (#2)*, $a$ = 8.76912(19) Å, $b$ = 9.1844(2) Å, $c$ = 16.8884(4) Å, $\alpha$ = 86.163(2)°, $\beta$ = 79.388(2)°, $\gamma$ = 82.3382(16)°, $V$ = 1323.72(6) Å$^3$, $T$ = −173(2) °C, $Z$ = 2, $\rho_{calcd.}$ = 1.677 g cm$^{-3}$,

$\mu$(Mo K$_\alpha$) = 3.123 mm$^{-1}$, $F(000)$ = 668, 6740 independent reflections, 5844 refined parameters, $R_1$ = 0.0485 [for $I > 2\sigma(I)$], $wR_2$ = 0.1071 (for all data), GOF = 1.186, and largest difference peak and hole: 0.59/−0.53 e Å$^{-3}$.

### 3.7.2. Crystallographic Data for **Z1Zn**

2(C$_{28}$H$_{20}$Cl$_4$N$_2$O$_3$Zn)·3(C$_4$H$_8$O), M$_r$ = 1495.66 g mol$^{-1}$, yellow block, monoclinic, space group *Cc* (#9), $a$ = 16.6298(3) Å, $b$ = 23.0060(4) Å, $c$ = 17.5585(3) Å, $\beta$ = 97.7800(15)°. $V$ = 6655.79(15) Å$^3$, $T$ = −173(2) °C, $Z$ = 4, $\rho_{calcd.}$ = 1.492 g cm$^{-3}$, $\mu$(Mo K$_\alpha$) = 1.102 mm$^{-1}$, $F(000)$ = 3072, 14627 independent reflections, 836 refined parameters, $R_1$ = 0.0385 [for $I > 2\sigma(I)$], $wR_2$ = 0.1037 (for all data), GOF = 1.019, and largest difference peak and hole: 1.40/−0.89 e Å$^{-3}$. Flack parameter: 0.073(3).

### 3.7.3. Crystallographic Data for **Z2Zn**

C$_{28}$H$_{22}$N$_4$O$_7$Zn, M$_r$ = 591.88 g mol$^{-1}$, yellow block, triclinic, space group *P*-1 (#2), $a$ = 11.6582(3) Å, $b$ = 11.7904(2) Å, $c$ = 20.2835(4) Å, $\alpha$ = 94.4710(16)°, $\beta$ = 102.436(2)°, $\gamma$ = 90.9990(18)°, $V$ = 2712.69(11) Å$^3$, $T$ = −173(2) °C, $Z$ = 4, $\rho_{calcd.}$ = 1.449 g cm$^{-3}$, $\mu$(Mo K$_\alpha$) = 0.959 mm$^{-1}$, $F(000)$ = 1216, 11549 independent reflections, 737 refined parameters, $R_1$ = 0.0543 [for $I > 2\sigma(I)$], $wR_2$ = 0.1230 (for all data), GOF = 1.078, and largest difference peak and hole: 1.20/−0.47 e Å$^{-3}$.

## 4. Conclusions

New Schiff bases and their Fe, Cu and Zn complexes were synthesized and characterized by various spectroscopic methods. The structure of the **ZH4**, **Z1Zn** and **Z2Zn** were determined by DFT calculations and X-ray diffraction measurements. The crystallographically determined geometry of the ligands, as well as the measured vibrational spectra of **ZH4**, **Z1Zn** and **Z2Zn,** are in good agreement with the theoretical results. The antimicrobial and anticancer activities of the Schiff bases and their metal complexes were evaluated. The compounds **ZH2–ZH4** and their zinc complexes showed antimicrobial activity against *Micrococcus luteus*, while **ZH2**, **ZH3**, **Z2Zn**, **Z3Zn** and **Z2Fe** showed antimicrobial activity against *Staphylococcus aureus*. An anticancer assay showed that most of our tested compounds had variable reduction in viability of the tested cancer cell lines compared to the normal fibroblasts. Moreover, **ZH1**, **Z2Fe** and all zinc complexes showed selective anticancer effects against the MCF7 cell line, and **Z2Cu** showed the lowest IC$_{50}$ values against all cell lines.

**Author Contributions:** Conceptualization, M.E.-k., T.M.A.A.-S. and H.K.; methodology, S.S.A.-T.; software, T.S.A.; validation, T.M.A.J., A.A.S. and Z.H.O.; formal analysis, M.S.A.Z. and W.A.; investigation, T.Q.; resources, T.M.A.J.; data curation, T.M.A.A.-S.; writing—original draft preparation, Y.Y., H.M. and M.E.-k.; writing—review and editing, H.M., H.K., Y.Y. and M.E.-k.; visualization, S.S.A.-T.; supervision, T.M.A.J.; project administration, T.M.A.A.-S. and T.S.A.; funding acquisition, T.M.A.J. All authors have read and agreed to the published version of the manuscript.

**Funding:** This research was funded by the Scientific Research Deanship at Yarmouk University (Irbid, Jordan), grant number 86/2021.

**Institutional Review Board Statement:** Not applicable.

**Informed Consent Statement:** Not applicable.

**Data Availability Statement:** The crystallographic data reported in this manuscript have been deposited with the Cambridge Crystallographic Data Centre under deposition no. CCDC-2141222, 2173302 and 2173303 for **ZH4**, **Z1Zn** and **Z2Zn**, respectively. This data can be obtained free of charge via http://www.ccdc.cam.ac.uk/data_request/cif (or from the Cambridge Crystallographic Data Centre, 12, Union Road, Cambridge, CB2 1EZ, UK).

**Acknowledgments:** We highly appreciate the generous financial support of project 86/2021 by the Scientific Research Deanship at Yarmouk University (Irbid, Jordan).

**Conflicts of Interest:** The authors declare no conflict of interest. The funders had no role in the design of the study; in the collection, analyses, or interpretation of data; in the writing of the manuscript or in the decision to publish the results.

**Sample Availability:** Samples of the compounds are available from the authors.

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
