# Peer review of "Synthesis, Characterization, Computational and Biological Activity of Some Schiff Bases and Their Fe, Cu and Zn Complexes"

_inorganics, doi:10.3390/inorganics10080112_

Round 1

Reviewer 1 Report

The paper "Synthesis, characterization, computational and biological activity of some Schiff bases and their Fe, Cu and Zn complexes" is somewhat standard, but not in a bad sense, report on the synthesis, structure, spectral properties and biological activity of four new ligands of Schiff base nature and their iron, copper and zinc complexes. The structure of both free ligands and metal complexes is rather twisted and complicated as it is shown by X-Ray data. Authors proved that metal ions are bound by two nitrogens and two oxygens of ligands. They demonstrated the ability of complexes to suppress the growth of some microorgamisms and cause the death of the tumor cells. The paper is well-written. I have, however, a couple of comments and questions:

1. Why the NMR data are reported for zinc complexes only? Did you encounter a strong paramagnetic shift by copper or iron ions or decided that it even does not worth a try? Crystal structure gives an idea that the central ion might be shielded efficiently from the "outer" molecule, it would be interesting to see if it is true by 1H NMR spectroscopy.

2. I recommend register 15N NMR spectra for one ligand and one zinc complex. It can be done without 15N enriching of the compounds though the time of experiment will be long (something like 16-20 h). The 2D 1H, 15N HMBC techniques is pretty efficient for registering the nitrogen spectra of 15N-poor samples. It will show directly the extremely strong displacement of nitrogen chemical shift upon complexation.

3. 10 mg per 1 mL is 10 g per L - pretty concentrated solution. Because of that, I have some doubts that the synthesized compounds are that good antibacterial agents. Moreover, there is, unfortunately, neither standard antibiotic nor anti-cancer compound for control used for comparison.

4. Moreover, DMSO itself can possess some anti-bacterial properties. The blank experiment with pure solvent must be performed to measure its inhibition zone of bacterial growth.

Therefore, I, unfortunately, have to recommend a major revision because the control is clearly missing in biological experiments. Otherwise it is a very good paper.

Author Response

Thank you very much for your important comments.

Reviewer 2 Report

The paper of Taghreed M.A. Jazzazi and co-authors is an interesting fundamental work on synthesis new Schiff bases (4 new Ligands) and their metal complexes (Zn, Cu(II), Fe(II)). Free organic ligands and Zn complexes were characterized with NMR and other methods, for one ligand and two Zn compounds were grown single crystals with a structure determination. There is also a biological part in the article with some interesting results. The current work seems interesting to me and I recommend it to publish in Inorganics.

But I have questions:

New organic Schiff bases and Zn compounds were characterized with NMR – to that part of the manuscript I don’t have questions. But it’s not enough IR, UV, CHN to prove the synthesis of Fe(II) and Cu(II) complexes.

For example, isostructural complexes of zinc, iron, and copper can be characterized by XRPD provided that the structure in a single crystal has been determined for the zinc complex – but here we have a lack of such information.

Copper(II) complexes are easily described using electron paramagnetic resonance (if single crystals cannot be isolated, this should be enough) – but here we have a lack of such information.

It is very interesting why the authors are sure that iron(II) acetate, which is extremely unstable in air in solution (or even solid state), does not turn into iron(III) acetate - the experimental part does not contain information about the inert atmosphere and Schlenk line. It is also not clear why should take absolute ethanol if the authors work with metal(II) acetate hydrates. There is not enough data on the magnetic susceptibility of the Iron(II) compound or the Mössbauer spectrum to say which state of the iron authors have.

I recommend doing the following: remove information on the synthesis of iron(II) and copper complexes, leaving only data on the synthesis of Schiff bases and their zinc complexes. Paramagnetic complexes should be further investigated according to given recommendations and a new article should be written on them.

Author Response

(The authors gave the same response as above.)

Reviewer 3 Report

I consider the subject of the manuscript entitled “Synthesis, characterization, computational and biological activity of some Schiff bases and their Fe, Cu and Zn complexes” to be of very high interest and very appropriate for the Inorganics journal. The synthesis and the investigation of the biological properties of transition metal complexes is extremely promising in terms of drug design, especially antitumor and antibacterial agents. The work presented in this manuscript is of high quality, the phisico-chemical and biological characterization of the newly synthesized ligand and its Zn(II), Fe(II) and Cu(II) complexes is thorough and reveals that the compounds may have therapeutic potential. 

Considering all the facts mentioned, I recommend publication with a few minor revisions detailed below.

Scheme 1 – the structure of the diamine should be revised, so that the two NH2 groups do not overlap

Line 98 – “may be attributed”

From Line 122 - Section “2.2. X-ray structural analysis” -  for the structural analysis of pentacoordinate metal complexes, we suggest using the parameter that Addison et al proposed, τ, as an index of the degree of trigonality, within the structural possibilities between the ideal trigonal bipyramidal and square pyramidal geometries. The geometric parameter was defined as τ = (β – α)/60, where α and β are the largest angles formed with the metal centre in the coordination polyhedron. For a perfect tetragonal geometry, τ = 0, while for an ideal trigonal bipyramide, τ = 1. (Addison AW, Rao TN, Reedijk J, van Rijn J, Verschoor GC. Synthesis, structure, and spectroscopic properties of copper(II) compounds containing nitrogen and sulphur donor ligands; the crystal and molecular structure of aqua[1,7-bis(N-methylbenzimidazol-2?-yl)-2,6-dithiaheptane]copper(II) perchlorate. J Chem Soc Dalton Trans. 1984 (7):1349)

From Line 231 – Table 5 – this table should bear the no. 2, as it is the second one to appear in the manuscript (it is also referred to as Table 2 in the text). Also, for a better understanding of the data presented, the specification “Inhibition zone (mm)” should appear for columns 2, 3 and 4 (the studied bacterial species)

From Line 235 – Figure 6 – on the second picture, a bacterial species appears, Serratia marcescens, is shown, but this is not mentioned anywhere else in the manuscript. On the other hand, the pictures do not show the results for Micrococcus luteus, but this species is mentioned in both Experimental and Results sections

Lines 243 – 244 – The human dermal fibroblasts cell line is once abbreviated “FDH” and once “HDF”. For a better understanding of the data, we suggest using the abbreviation “HDF” throughout the manuscript

From Line 248 – Table 3 – the data is not shown properly, perhaps the font should be a little bit smaller so that the data would fit

Lines 306 – 307 – was the metal acetate also dissolved? That is suggested by the expression “was added dropwise”, but no solvent is mentioned

Lines 439 – 440 – there is no reference to magnetic measurements in the manuscript

Author Response

(The authors gave the same response as above.)

Round 2

Reviewer 1 Report

Authors improved the manuscript and answered all the questions and comments. I recommend publishing the manuscript in its current form.

Reviewer 2 Report

The authors answered my questions, made minor changes to the manuscript. I recommended dividing the manuscript into two parts, finalizing the second part, publishing the results in another article. The authors did not do this, but promised. Despite this, there is enough data for publication in Inorganics.

It remains to wish good luck and I hope to see another article with diffraction patterns, EPR and X-ray diffraction analysis for copper and iron complexes.